# Differences in Microbial Community and Metabolites in Litter Layer of Plantation and Original Korean Pine Forests in North Temperate Zone

**DOI:** 10.3390/microorganisms8122023

**Published:** 2020-12-17

**Authors:** Yue Wang, Ting Li, Chongwei Li, Fuqiang Song

**Affiliations:** 1Engineering Research Center of Agricultural Microbiology Technology, Ministry of Education, Heilongjiang University, Harbin 150080, China; 2181167@s.hlju.edu.cn (Y.W.); 2191218@s.hlju.edu.cn (T.L.); 2Heilongjiang Provincial Key Laboratory of Ecological Restoration and Resource Utilization, College of Life Science, Heilongjiang University, Harbin 150080, China

**Keywords:** forest-litter layer, microbial-community structure, indicator bacteria, differential metabolites

## Abstract

In order to explore the relationship between microbial diversity and metabolites in the litter layer of northern temperate forests, the microbial community structure and metabolite species in the litter layer of an original Korean pine forest and Korean pine plantation of northern temperate climate were determined on the basis of high-throughput sequencing and metabonomic techniques. The results showed that there were 698 bacterial genera and 363 fungal genera in the litter samples in the original Korean pine forest. Linear discriminant effect size (LEfSe) analysis showed that there were 35 indicator bacterial species and 19 indicator fungal species. In the litter samples of the Korean pine plantation, there were 622 bacterial genera and 343 fungal genera. Additionally, LEfSe analysis showed that there were 18 indicator bacterial species and 5 indicator fungal species. The litter of the two forest types contained 285 kinds of organic compounds, among which 16 different metabolites were screened, including 6 kinds of organic acids, 5 kinds of amino acids, 2 kinds of sugars, 2 kinds of sugar alcohols, and 1 kind of lipid. Latescibacteria, Rokubacteria, and Olpidiomycota are unique to the original Korean pine forest. They can catalyze the degradation rate of litter and decompose cellulose and chitin, respectively. Subgroup 6 was abundant in the lower litter layer. Subgroup 6 can grow with carbon compounds as substrate. It was clear that the microbial diversity of the litter layer in the original Korean pine forest was higher than that of the Korean pine plantation. Moreover, whether original forest or plantation forest, the lower-litter layer microbial diversity was higher than that in the middle-litter layer. CCA showed that the main metabolites were related to *Chitinophagaceae_uncultured* were saccharopine. The main metabolites associated with *Mortierella* and *Polyscytalum* were myo-inositol. At the same time, analysis of the difference between the litter layer of the original Korean pine forest and the Korean pine plantation also provides a theoretical basis for their participation in the element cycles of forest ecosystems.

## 1. Introduction

Litter is an important component of the net primary productivity of terrestrial ecosystems, of which the yield and composition affect the material cycle of the ecosystem [1]. The material cycle of forest ecosystems mainly depends on the balance between plant growth and litter decomposition [2]. The main component of forest topsoil is made up of different parts of plant material, which falls on the soil surface and is immediately covered by various microorganisms; thus, the decomposition process begins [3,4]. As a unique structural layer of forest ecosystems, the litter layer plays an important role in forest soil development, carbon and nitrogen cycles, and energy flow [5,6]. There is a close relationship between the material cycle and the microorganisms and produced metabolites in forest litters. With the decline and fall of plant tissue, tissue in the form of litter is in the saprophytic process, so a variety of microorganisms in the saprophytic system participate in litter decomposition. In the process, they produce different metabolites accompanied by the degradation of litter into the forest soil environment, maintaining forest ecosystem carbon balance and nutrient cycling [7,8].

With the participation of microorganisms, humus such as litter is gradually decomposed into soil nutrients. Therefore, microorganisms, as a biological factor, play an important role in forest health and environmental adaptability in terms of maintaining soil structure, material transformation, nutrient cycling, and other aspects [9,10,11,12]. From central and southern Ural spruce and birch forest litter, in terms of microbes with metabolic characteristics, some researchers have found that spruce forest litter Acidobacteria numbers are lower than those in birch forest litter. Actinomyces and fungi of two forest types had decreased in alpha diversity, but beta diversity had increased. The forest litter layer first metabolizes a large number of water-soluble compounds, which provide carbon and nutrients for lower soil microorganisms [13]. In addition, the available carbon and nitrogen content of deciduous broad-leaf litter leachate is higher than that of coniferous litter, which can increase the respiration of soil microorganisms and lead to higher carbon-utilization efficiency of microbial communities [14]. Studies on the changes of fungal communities in litter decomposition for more than two years in the boreal forest of Alaska have shown that the fungal community structure changed with time and litter type and that the metabolite glutamate and the tannin–protein complex were more common in the late decomposition stage [15]. Different microbial species can play different roles in forest litter.

Metabonomics is the large-scale study of low-molecular-weight organic compounds in soil. It provides a method to characterize soil and assess the metabolic status of soil biomes [16,17,18] and to predict an important part of systems biology; this method has developed rapidly and penetrated many research fields in recent years. Forest litter layers have large metabolite systems because of their high environmental complexity [19]. Studies have found that the decomposition effect of litters is closely related to the soil–nutrient cycle. Polyphenols are important secondary metabolites of litter layers and a major obstacle to the partial decomposition of plant leaves [20]. The characteristics of litter and soil phospholipid fatty acid (PLFA) of four forest types in Germany, within the same season for three consecutive years, were studied. The litter layer was more susceptible to the influence of drying, and its total PLFA was decreased by nearly 30% in a dry year [21]. At present, some researchers have pointed out that forest litters have an important relationship with the activity of lignin-degrading enzymes in the decomposition stage [22], and these enzymes can regulate the degree of litter decay [23]. In a temperate deciduous Pennsylvania forest, researchers studying the forest litter and soil metabolites on the surface using biomarker analysis and nuclear-magnetic-resonance spectrum technology found biomarkers of suberin and lignin rather than free sugars and the free-ring type of lipid reserves from the litter to the surface soil, which was free of acyclic lipid and cutin [24]. Microorganisms and their metabolites play a vital role in the material cycle of forest litter.

Previous studies have mainly focused on identifying specific microorganisms in forest litter, but only a few studies have focused on the secondary metabolites of forest litters. At present, studies on the microbial and metabolic functions of litter in northern temperate forests are still lacking. We hypothesize that (1) the microbial diversity of original Korean pine forests is higher than that of a Korean pine plantation in the decomposition process of forest litter, and (2) the decomposition rate of forest litter is closely related to microbial community composition and the related metabolites.

## 2. Materials and Methods

### 2.1. Natural Survey of Study Area

Zonal vegetation in the Liangshui Nature Reserve is a temperate coniferous broad-leaved mixed forest dominated by Korean pine that belongs to the northern subzone of the temperate coniferous broad-leaved mixed-forest zone. The typical broad-leaved coniferous forest subzone is the central zone of the distribution of the world’s coniferous forests, which has great typicality and representativity. The Liangshui National Nature Reserve is located in the Dailing district, Yichun city, Heilongjiang province. Its geographical coordinates are 128°53′20″ E and 47°10′50″ N. It is a typical low, mountainous, and hilly landform, with obvious temperate continental monsoon climate characteristics and a total forest reserve of 1.7 million m^3^. Annual average temperature is −0.3 °C, annual average precipitation time is 120 days, annual average precipitation is 676 mm, annual average evaporation is 805 mm, the annual sunshine time is 1850 h, daily light rate is 43.5%, winters are cold and long, and frozen-soil depth can reach about 2.0 m.

### 2.2. Sample Collection

Forest litter is composed of dead branches, leaves, bark, herbs, and other biological residues. The study on the Liangshui Nature Reserve examined the half decomposition of the forest-litter layer’s leaf mesophyll tissue (gray to black, quality of material softening, beginning to rot, recognizing the shape of the leaves, the part without a complete leaf shape, but well-recognized coniferous or broad leaves, gray–black or dark brown) and decomposition layer (also called the humus layer; original litter shape completely unidentifiable, finely powdered dark brown rather than mineral soil, soft and elastic). Introduction of terraced fields are shown in Table 1. All samples in this study were collected in the freezing and thawing period at the end of September 2019 in the Liangshui National Nature Reserve that is adjacent to the same latitude of the original Korean pine forest and the Korean pine plantation. Three plots were selected from the original forest and the plantation forest, respectively (samples from each plot were sampled by the five-point sampling method). Each plot was 1 km apart. There were two forest types investigated—plantation forest (A) and original forest (B)—within two litter layers, namely, middle (X) and lower layer (Z). In order to prevent the volatilization of the metabolites, samples were packed into sterilized self-sealing bags, placed in iceboxes, and brought back to the laboratory, where they were stored in liquid nitrogen at −80 °C.

### 2.3. Methods

#### 2.3.1. Extraction and Sequencing of Total DNA from Litter Layer

Total DNA of bacteria and fungi in the litter were extracted by a Fast DNA SPIN Kit (MP Biomedicals, CA, USA), detected by 1% agar gel electrophoresis, and purified by a Power Clean DNA Clean-Up Kit (MoBio, CA, USA); its concentration and purity were measured by Nanodrop 2000. The V3–V4 region of the 16S rRNA gene was amplified with universal bacterial primers 341F (5′-CCTAYGGGRBGCASCAG-3′) and 806R (5′-GGACTACNNGGGTATCTAAT-3′), and 18S rRNA was amplified with ITS1F (5′-CTTGGTCATTTAGAGGAAGTAA-3′) and IST2R (5′-GCTGCGTTCTTCATCGATGC-3′), a universal fungal primer. Bacterial amplification conditions were as follows: predenaturation at 98 °C for 1 min, denaturation at 98 °C for 10 s and cycling for 30 s, annealing at 50 °C for 30 s, extension at 72 °C for 30 s, and extension at 72 °C for 5 min. Fungal amplification conditions were as follows: predenaturation at 94 ° C for 5 min, denaturation at 94 °C for 30 min, renaturation at 60 °C for 30 s, extension at 72 °C for 15 s, 30 cycles, and, lastly, an extension at 72 °C for 10 min. QuantiFluor™-ST Blue fluorescence quantitative system (Promega) was used for the QuantiFluor™-ST quantitative assay after PCR amplification. Illumina Miseq Nano DNA LT Library Prep Kit was used to prepare the sequencing library. The Miseq platform was used for double-ended 250-bp sequencing. Barcode tag sequence and preprimer sequence were used to screen out valid sequences from the data. Sequencing was commissioned by Shanghai (China) Ling En Biotechnology Co. Ltd.

#### 2.3.2. Determination of Untargeted Metabolites in Litter Layer

First, the litter was thawed out and examined on ice. Briefly, 100 g of litter was taken, and 100 mL pure water (including internal standard) solution was added. The litter was vortexed for 2 min and centrifuged at 13,000 r/min for 10 min. Then, 5 mL supernatant was injected for nontargeted metabolome analysis. An ACQUITY UPLC HSS T3 column (100 × 2.1 mm, 1.7 m, Waters, Milford, MA, USA) was used for chromatographic separation. The chromatographic conditions were mobile phase A, water and 0.1% formic acid; mobile phase B, acetonitrile. The column temperature was 40 °C, and the flow rate was 0.35 mL/min. Gradient elution method: 0–1.0 min, 5% B; 1.0–9.0 min, 5–10% B; 9.0 to 12.0 min, 100% B; 12.0 to 15.0 min 5% B. Sample size was 5 L. Ion-source parameters: positive ion mode (sheath gas-flow rate, 40 ARB; auxiliary gas velocity, 10 ARB; spray voltage, 3.50 kV; ion-transport-tube temperature, 320 °C; ion-source temperature, 300 °C); anion mode (sheath gas-flow rate, 38 ARB; auxiliary gas velocity, 10 ARB; spray voltage, 3.00 kV; ion-transport-tube temperature, 320 °C; ion-source temperature, 300 °C).

### 2.4. Data Statistics

We used FLASH software filtering under the condition of certain filtering of the original data and quality control in accordance with QIIME V1.7.0 high-quality data. Sequence length <200 bp, average quality <25 or containing a small number of bases, effectively using the UCHIME algorithm and Gold database to chimerically detect and remove, after similarity, >97% of the sequences classified into the same operating classification unit (OTU) for each filtered OTU on behalf of the sequence and annotation. Lastly, each data sample was normalized.

Statistical software SPSS 21.0 was used for one-way ANOVA, and the *p* < 0.05 threshold value was used to characterize significant differences between the two groups of data. Differential metabolites were screened by using the variable-importance-in-projection (VIP) value of the pls-da model (threshold ≥1) and *p*-value (≤0.05) and fold change ≥2 of the independent-sample *t*-test to find differential metabolites. The qualitative method of differential metabolites was matched and screened with the METLIN online database. A nonparametric-factor Kruskal–Wallis (KW) sum-rank test was applied to identify significant taxa, and linear discriminant analysis (LDA) was applied to evaluate the effect of each feature. An LDA threshold of 3.0 and a significant *p*-value were used to detect biomarkers. Canonical correspondence analysis was performed using Canoco 5.

## 3. Results and Analysis

### 3.1. Bacterial Community Structure of Litter Layer in Korean Pine Plantation and Original Korean Pine Forest

#### 3.1.1. Results of Illumina-Miseq Sequencing of Litter-Layer Bacteria

A Venn diagram is used to show the number of OTUs contained in the four groups. There were a total of 4968 OTUs in the two forest types, among which 837 were specific to A and 1893 were specific to B (Figure 1a). There were 3422 OTUs in the middle and lower layers of the plantation forest, of which 1050 were AZ-unique and 1333 were AX-unique (Figure 1b). The number of core OTUs in the middle and lower layers of the original forest was 4825, among which 723 were BZ-specific and 1313 were BX-specific (Figure 1c). Principal component analysis of the bacterial community composition of the four groups of forest litter in the plantation forest and the original forest showed that the differential contribution value of the first component was 37.39%, the second component 17.863%, and the third component 17.024%. In terms of the three principal components, the bacteria of the plantation forest and the original forest could be significantly separated, indicating that the bacterial community structures of the plantation forest and the original forest were significantly different (Figure A1, Appendix A).

#### 3.1.2. α-Diversity of Bacterial Community in Litter Layer

Chao1, Shannon, and Simpson diversity indices were used to characterize the α-diversity of bacterial litter communities. As shown in Table 2, the Chao1 index in both types of forests showed increasing trends by litter layer. The BZ-treated litter’s bacterial-community richness index was 4853, which was significantly higher than that of AZ (*p* < 0.05). The highest index of BX was 5937, which was significantly higher than that of AZ (*p* < 0.05). The change of the Shannon index was consistent with the change trend of the Chao1 index. The bacterial-community richness of litter treated by BZ was 7.36, which was significantly higher than that of AZ (*p* < 0.05), while that of BX was as high as 7.53, which was significantly higher than that of AX (*p* < 0.05). The change trend of the Simpson index was contrary to that of the Chao1 and Shannon indices. BX had the lowest change trend value, 0.0012, and the highest richness, while AZ had the highest change trend value, 0.0049, and the lowest richness.

#### 3.1.3. Bacterial Community Composition in Litter Layer of the Two Forests

Bacteria that were identified in the litter layer of the Korean pine plantation included 28 phyla, 327 families, and 622 genera. Litter of the original Korean pine forest had 32 phyla, 358 families, and 698 genera identified. Species analysis was carried out at the phylum classification level (relative abundance >1%). Among the four groups of samples, Proteobacteria (18.16–26.19%) had the highest proportion, followed by Acidobacteria (11.14–13.84%), Bacteroidetes (5.86–7.21%), Verrucomicrobia (3.00–5.79%), Actinobacteria (1.99–2.22%), Patescibacteria (1.86–1.88%), and Gemmatimonadetes (1.32–1.56%) (Figure 2).

Figure 3a shows that there were 41 significant bacterial genus differences between the Korean pine plantation and the original Korean pine forest (*p* < 0.05), among which those of the *Rhodanobacter* microbes were the most significant (*p* < 0.001). Among them, *Rhodanobacter*, *Caulobacteraceae_unclutured*, *WD260_norank*, *Acetobacteraceae_uncultured*, and *Micropepsaceae_uncultured* were significantly higher in the plantation forest than those in the original forest. Subgroup 17_norank, *Blastocatellaceae_uncultured*, *Desulfarculaceae_uncultured*, *Stenotrophobacter*, and Subgroup 6_norank in the original forest were significantly higher than those in the plantation forest. According to Figure 3b, there were 31 bacterial genera in the middle and lower layers of the Korean pine plantation with significant differences (*p* < 0.05), among which that of the *Paucibacter* was the most significant (*p* < 0.001). Among them, *Paucibacter*, *Caulobacter*, and *Mycobacterium* were significantly higher in the middle layer than the lower layer. Figure 3c shows that there were a total of 35 bacteria genera with significant differences between the middle and lower layers of the original Korean pine forest (*p* < 0.05), of which those of *Rhizobacter* microbes were the most significant (*p* < 0.001). *Rhizobacter* was significantly higher in the middle layer than the lower layer.

#### 3.1.4. Bacterial Linear Discriminant Effect Size (LEfSe) Litter Analysis

LEfSe analysis can identify the bacterial community or species in each sample that is closely related to the metabolites with the greatest contribution. Original and plantation forests had 53 biomarkers (LDA score >3; Figure 4a). There were 35 groups of original forests; on the whole, the largest contribution to the original forests was from Subgroup_6, followed by *Blastolia_*Subgroup_4. There were 18 groups of plantation forests, of which *Alphaproteobacteria* microbes contributed the most, followed by *Micropepsales*, *Micropepsaceae*, and *Rhodanobacteraceae*.

There were 52 biomarkers (LDA score >3) in the middle and lower layers of the plantation forest (Figure 4b), and there were 24 groups in the middle layer. On the whole, *Proteobacteria* contributed the most to the middle layer of the plantation. This was followed by *Alphaproteobacteria*, *Sphingomonadales*, *Sphingomonadaceae*, *Bacteroidia*, *Bacteroides*, and *Caulobacteria*. There were 28 groups in the lower layer of the plantation forest, of which the *Acidobacteria*, *Solibacteraceae_*Subgroup_3, and *Solibacterales* contributed the most.

There were 34 biomarkers (LDA score >3) in the middle and lower levels of the original forest (Figure 4c), and there were 22 groups in the middle level of the original forest. On the whole, *Proteobacteria*, *Alphaproteobacteria*, and *Gammaproteobacteria* contributed the most to the middle level of the original forest. There were 12 groups in the lower layer of the original forest, of which *Acidobacteria* contributed the most, followed by Subgroup_6.

### 3.2. Community Structure of Litter-Layer Fungi in Plantation and Original Korean Pine Forests

#### 3.2.1. Results of Illumina-Miseq Sequencing of Litter-Layer Fungi

A Venn diagram is used to show the number of OTUs contained in the four forest litters. The total number of OTUs in the two types of forest litter was 680, among which 612 were specific to A and 820 were specific to B (Figure 5a). A total of 505 OTUs were found in the middle and lower layers of the plantation forest, among which 377 were AX-specific and 410 were AZ-specific OTUs (Figure 5b). There were 679 OTUs in the middle and lower layers of the original forest, among which 452 were BX-unique and 369 were BZ-unique OTUs (Figure 5c). Principal component analysis of the fungal-community composition of the four groups of forest litter in the plantation forest and the original forest showed that the contribution value of the first component of difference was 24.01%, the second was 13.8%, the third was 6.589%. Among them, forest distance between the three lower samples explains community composition similarity. Other samples were relatively dispersed, indicating large differences in OTU composition. In the first principal component, the middle and lower groups were significantly separated, indicating that the community structure of the middle and lower fungi was significantly different. In terms of the second principal component, the samples were not significantly separated for fungi. In terms of the third principal component, the two groups of original forest and plantation forest could be significantly separated, indicating that the community structure of the original forest and plantation forest was significantly different (Figure A2, Appendix A).

#### 3.2.2. α-Diversity of Litter-Layer Fungus Communities

The Chao1, Shannon, and Simpson diversity indices were used to characterize the α-diversity of fungal litter communities. As shown in Table 3, the Chao1 index showed that the AZ litter had the highest bacterial-community richness index at 634, which was significantly higher than that of BZ (*p* < 0.05), and AX was 496, which was significantly lower than that of BX (*p* < 0.05). The Shannon index showed that BZ reached a maximum of 4.34, which was significantly higher than AZ, AX, and BX (*p* < 0.05). The trend of the Simpson index was contrary to that of the Chao1 and Shannon indices. BZ had the lowest index of 0.0377 and the highest richness, while AX had the highest index of 0.0538 and the lowest richness.

#### 3.2.3. Composition of Litter-Layer Fungal Communities

The litter layer of the Korean pine plantation forest had 6 phyla, 163 families, and 343 genera of fungus identified. The litter layer of the original Korean pine forest had 6 phyla, 170 families, and 363 genera identified. Species analysis was performed at the phylum classification level (relative abundance >1%). Among the four groups of samples, Ascomycata was the highest (33.19–38.77%), followed by Mucoromycota (24.00–26.42%), unclassified phyla (14.57–15.00%), and Basidiomycota (6.24–11.30%) (Figure 6).

The abundance of dominant fungi in litter was analyzed at the genus level. According to Figure 7a, between the plantation forest and the original Korean pine forest, there were 12 fungal genera with significant differences (*p* < 0.05), among which *Colletotrichum* had the most significant difference (*p* < 0.001); the fungal genera of the original forest were significantly higher than those of the plantation forest. According to Figure 7b, there were 8 fungi genera in the middle and lower layers of the Korean pine plantation forest, with significant differences (*p* < 0.05), among which *Rhodotorula* had the most significant difference (*p* < 0.001); among them, *Rhodotorula* was significantly higher in the middle layer than in the lower layer, and *Pseudogymnoascus* was significantly higher in the lower layer than in the middle layer. According to Figure 7c, there were a total of 10 fungal genera in the middle and lower layers of the original Korean pine forest, with significant differences (*p* < 0.05), among which *Mortierella* had the most significant difference (*p* < 0.05), and the lower layer was higher in dominant fungi than the middle layer.

#### 3.2.4. LEfSe Analysis of Litter Fungi

There were 24 biomarkers (LDA score >3) for fungal LEfSe analysis in both the original and plantation forests (Figure 8a). There were 19 taxa in the original forest. On the whole, *Tetracladium* contributed the most to the original forest, while there were 5 taxa in the plantation forest. *Hypocreales* contributed the most to the plantation forest, followed by *Chloridium*.

LEfSe analysis had 19 biomarkers (LDA score >3; Figure 8b) in the litter layer of the plantation forest. There were 8 groups in the middle layer of the plantation forest. On the whole, *Ascomycota* contributed the most to the middle layer of the plantation forest, followed by *Xylariales*, *Xenopolyscytalum*, *Ophiostomataceae*, *Ophiostomatales*, *Graphilbum*, and *Polyscytalum.* There were 11 groups in the lower layer of the plantation forest, of which the largest contribution was from *Mucoromycota*, followed by *Agaricales*.

LEfSe analysis had 14 biomarkers (LDA score >3; Figure 8c) in the litter layer of the original forest. There were 9 groups in the middle layer of the original forest. On the whole, the largest contribution to the middle of the forest was from *Ascomycota*, followed by *Leotiomycetes*, *Sordariomycetes*, *Erysiphales*, *Erysiphaceae*, and *Golovinomyces*. There were 5 taxa in the lower layer of the original forest. The *Mortierella* genus contributed the most to the lower layer of the original forest, followed by *Mortierellales*, *Mortierellomycetes*, *Mortierellaceae*, *Mucoromycota*, *Sarocladiaceae*, and *Sarocladium*.

### 3.3. Litter-Layer Metabolomics of Korean Pine Plantation and Original Korean Pine Forest

The orthogonal projections of the latent structure discriminant analysis (OPLS–DA) model was established by using software-automated model-fitting analysis and replacement tests. The model showed good robustness and no overfitting phenomenon, and a PCA score graph was drawn. Samples were in a significantly dispersed state, so there was a significant difference in the metabolic status of the samples. The distribution of the samples in the plantation forest and original forest groups showed a certain degree of differentiation, and the samples had a uniform degree of differentiation, which allowed for the next differential metabolite screening (Figure 9).

#### 3.3.1. Screening of Different Litter Metabolites

A total of 285 compounds were detected in the litter layer by LC–MS, including sugars, amino acids, organic acids, nucleobases, sugar alcohols, lipids, and a range of secondary metabolites. According to the VIP value of the OPLS–DA model (threshold ≥1), fold change ≥2 was used to look for metabolites with different expressions by combining the *p*-value (*p*-value ≤ 0.05) of the independent sample *t*-test, as shown in Table 4.

#### 3.3.2. Differential-Metabolite-Cluster (Heatmap) Analysis of Litter

Hierarchical cluster analysis of the above characteristic metabolites showed that 16 (Figure 10a), 41 (Figure 10b), and 21 (Figure 10c) different clusters were generated from the samples. The contents of dodecanoic acid, saccharopine, phosphohydroxypyruvic acid, myo-inositol, D-galactose, costunolide, 19(S)-HETE, palmitoleic acid, and dimethylglycine in the Korean pine plantation forest were higher and showed a significant upward trend. In the original Korean pine forest, sanguinarine, 5-dehydroepisterol, d-fructose, d-piperideine-2-carboxyli, carboxyli, pyroglutamic acid, and 2-aminophenol were all higher in content, showing a significant upward trend.

In the Korean pine plantation forest, contents of mid-layer litter (S)—norlaudanine, sakuranetin, kaempferide, apigenin, and lanosterin—were higher, showing a significant upregulated trend. The content of geranylgeraniol in the lower litter layer was higher, showing a significant upregulated trend. In the original Korean pine forest, contents of sakuranetin, apigenin, tryptamine, and other litter in the middle layer were higher, showing a significant upregulated trend.

### 3.4. Canonical Correspondence Analysis (CCA) of Relative Microbial Abundance and Differential Metabolites

CCA was performed on differential metabolites from 28 dominant bacterial genera and types of litter (Figure 11a). The cumulative interpretation rate of the first sequence axis was 41.70%, that of the second sequence axis was 74.14%, and that of the two axes was 115.84%. Saccharopine was the main influencing factor, and its explanatory rate was 23.8%, followed by pyroglutamic acid (15.8%). In addition, saccharopine, d-fructose, and dodecanoic acid were significantly correlated with the relative abundance of *Chitinophagaceae_uncultured, SC-I-84_norank,* and *Haliangium,* respectively (*p* < 0.05).

CCA was performed on differential metabolites between 25 dominant fungal genera and types of litter (Figure 11b). The cumulative interpretation rate of the first ordination axis was 26.42%, the second ordination axis was 45.85%, and the total cumulative interpretation rate of the two axes was 72.27%. Myo-inositol was the main influencing factor, and its interpretation rate was 17.1%, followed by costunolide (interpretation rate, 15.4%) and d-galactose (interpretation rate, 15.1%). In addition, myo-inositol was significantly correlated with the relative abundance of *Lactarius* (*p* < 0.05). Costunolide was significantly correlated with the relative abundance of *Graphilbum and Mytilinidion* (*p* < 0.05). d-Galactose was significantly correlated with the relative abundance of *Polyscytalum and Sporidesmium* (*p* < 0.05).

## 4. Discussion

### 4.1. Effects of Bacterial Community and Metabolites on Forest-Litter Decomposition

Chao1, Shannon, and Simpson diversity indices were used to characterize the α diversity of bacterial litter communities (Table 2). The Chao1 and Shannon indices showed that the litter bacterial communities in the original forest were significantly higher than that of the plantation forest (*p* < 0.05). The lower-layer litter bacterial communities in the original forest were significantly higher than that of the original forest (*p* < 0.05).

The structure of the bacterial microbial community indicated that Subgroup_6 of Acidobacteria had the greatest contribution in the original forest and the lower layer of the plantation forest (Figure 4). Subgroup_6 of Acidobacteria can grow on the substrate of litter-layer carbohydrates and participate in carbohydrate metabolism [25]; d-fructose content in the litter showed a significant upregulation trend (Figure 10). Combined with CCA results (Figure 11), d-fructose content was significantly correlated with the relative abundance of the dominant bacteria; it was speculated that Subgroup_6 of Acidobacteria could transform multiple carbohydrates into monosaccharides in the process of forest-litter metabolism. Proteobacteria had the greatest contribution to the middle layer of litter, which was related to the high hemoglobin content in the middle litter layer. Proteobacteria can use hydropyrite as an electron acceptor to drive iron-ore reduction under anoxic environments [26] and the reduction of iron-containing organic matter to ferrous atoms, resulting in an increase in heme content. In addition, LEfSe analysis showed that Alphaproteobacteria also had a relatively high contribution to the litter layer (Figure 4). As Alphaproteobacteria are a major bacterium that degrades biofilm and has a strong corrosion rate to waste [27], we hypothesize that they play a critical role in litter degradation.

Different to the plantation forest, Latescibacteria and Rokubacteria were also found in the litter layer of the original forest. Studies have shown that Latescibacteria belong to saprophytic organisms and have a significant ability to degrade proteins, lipids, and polysaccharides in plant and microbial cell walls [28] to accelerate the decomposition and utilization of materials. There is strong genomic heterogeneity between cells in Rokubacteria, which are capable of mixed metabolism using a variety of nutrients [29].

### 4.2. Effects of Fungal Communities and Metabolites on Forest-Litter Decomposition

Chao1, Shannon, and Simpson diversity indices were used to characterize the α diversity of fungal litter communities (Table 3). The Chao1 index showed that the middle litter-layer fungal communities in the Korean pine plantation forest had the highest richness. The Shannon index showed that the middle layer of the plantation forest had the highest richness. The variation trend of the Simpson index was contrary to those of the Chao1 and Shannon indices, which indicates that the middle-layer richness of the original Korean pine forest was the highest, while the lower-layer richness of the Korean pine plantation forest was the lowest.

The fungal microbial-community structure indicated that Ascomycota had the highest abundance in the litter layer and made the greatest contribution to the middle layer of both forest types of litter (Figure 8). Ascomycetes have strong metabolic activity, and the metabolites include terpenes, alkaloids, flavonoids, and sterols [30]. Combined with the CCA results, *Ascomycetes* made the greatest contribution to metabolites such as myo-inositol and costunolide. Therefore, we infer that the content of metabolites in the middle layer of litter is higher than that in the lower layer, which should be the role of *Ascomycetes*. *Hypocreales* contribute the most to the litter layer of plantation forests, breaking down cellulose and other organic matter [31]. d-galactose content in the plantation forest showed a significant upregulation trend, and CCA results showed that d-galactose content was significantly correlated with fungal abundance (Figure 11), suggesting that Candida decomposes organic matter, leading to an increase in d-galactose content. LEfSe analysis showed that the middle-layer contribution of the two types of litter was followed by *Leotiomycetes*, while *Leotiomycetes* existed as the dominant bacterium, suggesting that, although the metabolic yield of the middle layer of litter was higher than that of the lower layer, the middle layer was relatively poor in nutrition because there were more *Leotiomycetes* in the barren ecosystem [32]. The abundance of *Mortierella* was highest in both the plantation forest and the lower part of the original forest (Figure 8). Studies have shown that *Mortierella* has an obvious effect on the prevention and control of Meloidogyne in vitro. *Mortierella* can adhere to the stratum corneum through hyphae, penetrate the stratum corneum, and then digest substances in its cells, thereby reducing Meloidogyne damage to plants and protecting the forest ecosystem [33] as a natural barrier for the ecosystem of the Liangshui Nature Reserve.

The litter in the original forest contained a class of fungi not found in the plantation forest, Olpidiomycota. Studies showed that Olpidiomycota have a dual role. On the one hand, infecting the host, leading to plant infection with brown spot disease; on the other hand, most species can break down cellulose and chitin [34] easily to provide more nutrients for plant growth and promote the material circulation of the ecosystem. Therefore, although some microorganisms are not dominant in forest litter, their role cannot be ignored and they play a crucial role in the decomposition of forest litter. In the future degradation of litter and other substances, we can scientifically utilize these micro-but-prominent forest terrestrial environmental microorganisms to increase the nutrient cycling of terrestrial forest environments.

Due to limitations of the metabolite database, metabolites detected in this study may not be complete, and undetected metabolites may be more meaningful to understand the function of soil microorganisms. So, as future research direction, by using targeted mass spectrometry combined with nuclear magnetic resonance (NMR), we can explore unknown metabolite structures and properties, complement the metabonomics research of forest litter layers, or combine genomics, proteomics, and transcriptome studies to clarify the metabolite-induced genomic or proteomic regulatory mechanism.

## 5. Conclusions

By analyzing the microbial community structure and metabolite types at key layers of litter in original and plantation Korean pine forests, it was found that the microbial diversity of the original Korean pine forest was higher than that of the Korean pine plantation forest during the decomposition of litter. At the bacterial species level, the main indicator bacteria of the original Korean pine forest were Subgroup_6 and *Blastocatellia_*Subgroup_4, and the main indicator bacteria of the Korean pine plantation forest were *Alphaproteobactria*, *Micropepsales*, *Micropepsaceae*, and *Rhodanobacteraceae*. At the fungal species level, *Tetracladium* was the main indicator of the original Korean pine forest, and *Hypocreales* and *Chloridium* were the main indicators of the Korean pine plantation forest. CCA showed that the main metabolite related to *Chitinophagaceae_uncultured* was saccharopine. The main metabolite associated with *Mortierella and Polyscytalum* was myo-inositol.

## Figures and Tables

**Figure 1 microorganisms-08-02023-f001:**
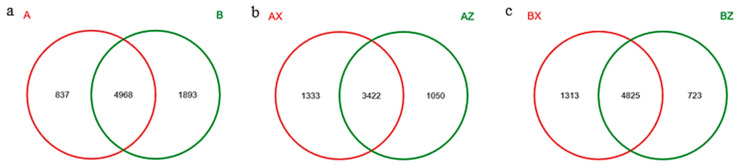
Venn diagram sample analysis. (**a**) Plantation forest (A) and original forest (B), showing the OTU numbers of bacteria; (**b**) middle litter layers in plantation forest (AZ) and lower litter layers in plantation forest (AX), showing the OTU numbers of bacteria; (**c**) middle litter layers in original forest (BZ) and lower litter layers in original forest (BX), showing the OTU numbers of bacteria.

**Figure 2 microorganisms-08-02023-f002:**
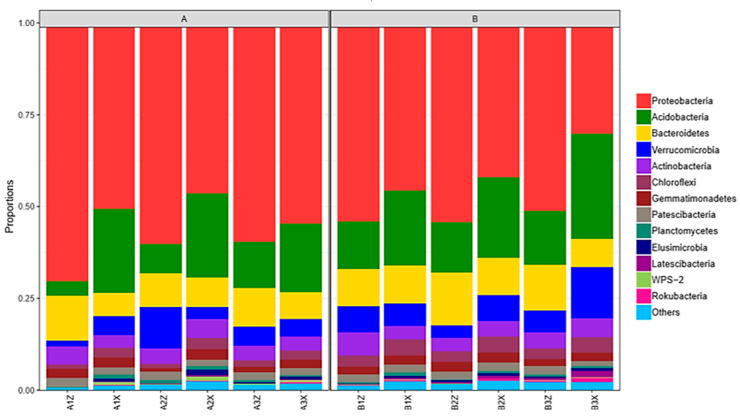
Bacterial species structure of plantation forest litter (**A**) and original forest litter (**B**) at the phylum classification level.

**Figure 3 microorganisms-08-02023-f003:**
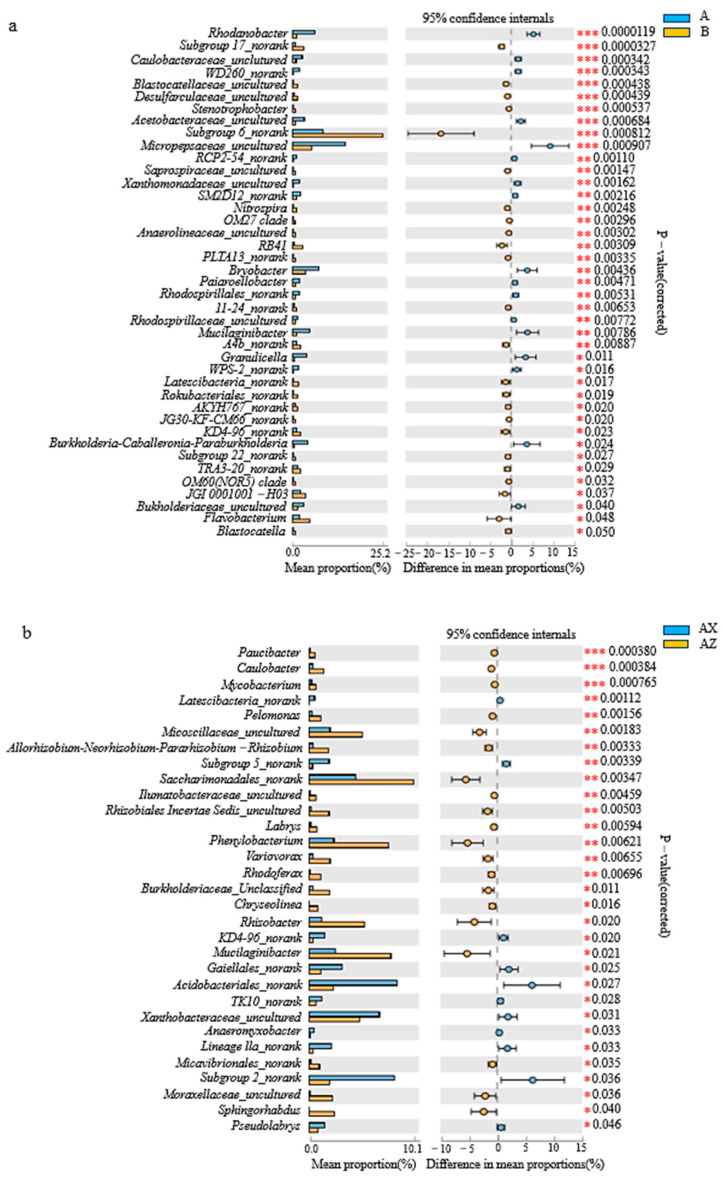
Differential analysis of the horizontal abundance of bacterial genera in litter. (**a**) Difference in bacterial abundance between litter layer of the Korean pine plantation and the original Korean pine forest; (**b**) different bacterial genera between the middle and lower litter layers in the Korean pine plantation; (**c**) different bacterial genera between middle and lower litter layers in the original Korean pine forest. *, significant difference between the two groups of data (*n* = 3, *p* < 0.05); **, very significant difference between the two groups of data (*n* = 3, *p* < 0.01); ***, very significant difference between the two groups (*n* = 3, *p* < 0.001).

**Figure 4 microorganisms-08-02023-f004:**
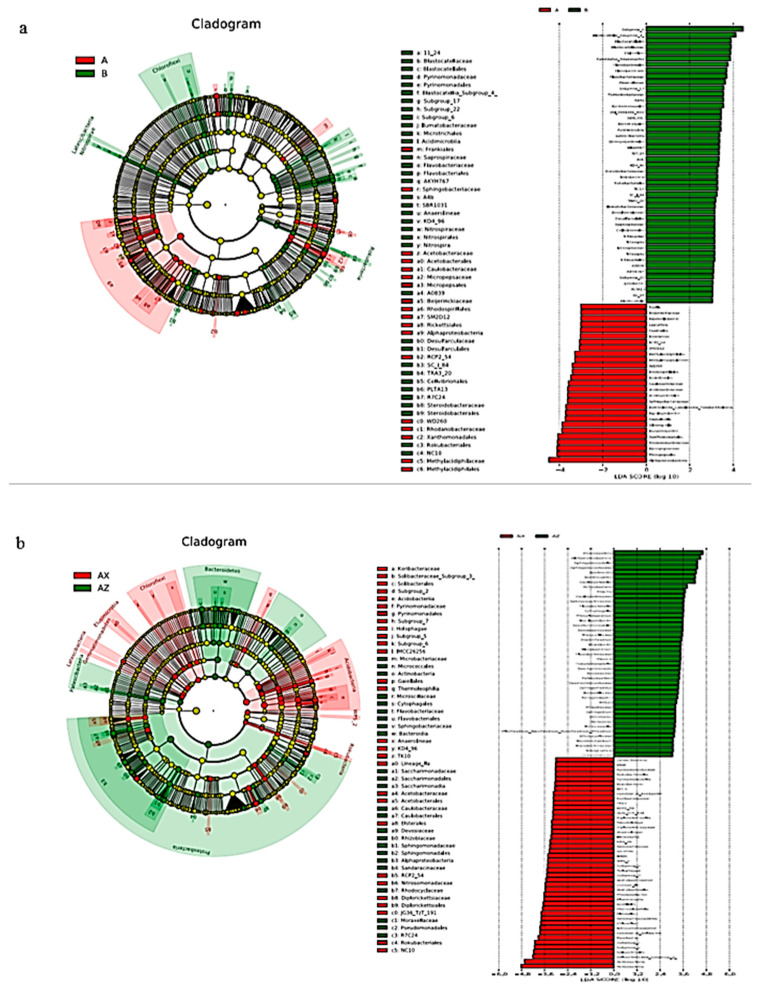
Bacterial linear discriminant effect size (LEfSe) analysis of litter layers. Cluster tree of bacterial LEfSe analysis in (**a**) Korean pine plantation and original Korean pine forest, (**b**) different litter layers in Korean pine plantation, and (**c**) different litter layers of original Korean pine forest.

**Figure 5 microorganisms-08-02023-f005:**
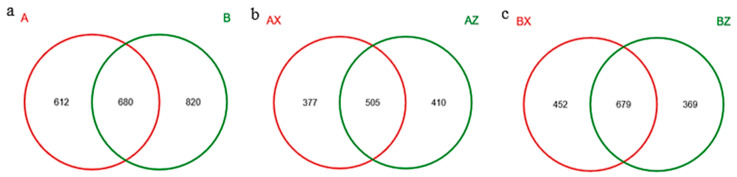
Venn diagram sample analysis on operating classification unit (OTU) number. (**a**) Plantation forest (A) and original forest (B), showing the OTU numbers of fungi; (**b**) middle litter layers in plantation forest (AZ) and lower litter layers in plantation forest (AX), showing the OTU numbers of fungi; (**c**) middle litter layers in original forest (BZ) and lower litter layers in original forest (BX), showing the OTU numbers of fungi.

**Figure 6 microorganisms-08-02023-f006:**
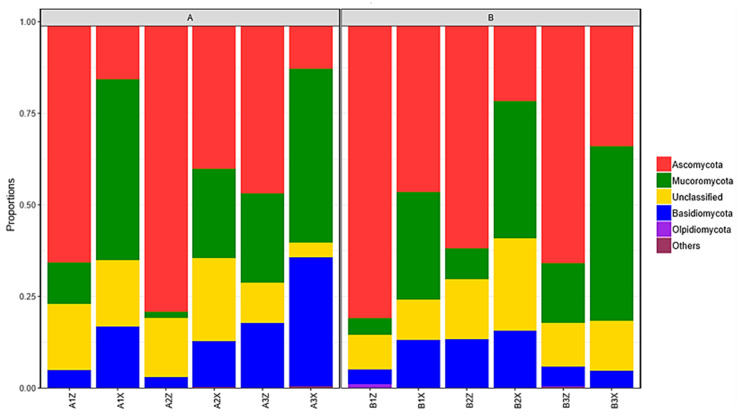
Fungal species structures of plantation forest litter (**A**) and original forest litter (**B**) at the phylum taxonomic level.

**Figure 7 microorganisms-08-02023-f007:**
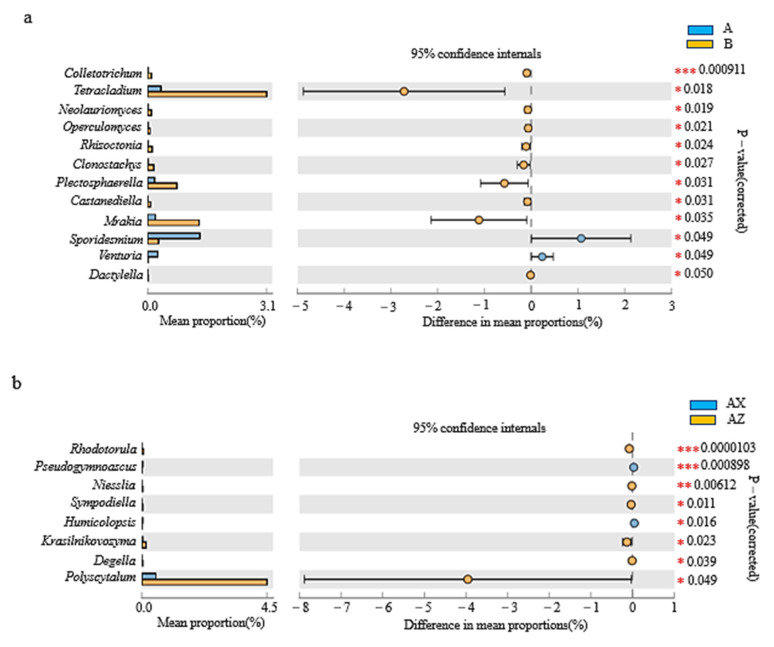
Differential analysis of horizontal fungus abundance in litter. Difference of fungal abundance between (**a**) litter layers of plantation forest and original Korean pine forests, (**b**) middle layer and lower litter layers in Korean pine plantation forest, and (**c**) middle and lower litter layers in original Korean pine forest. *, significant difference between the two groups of data (*n* = 3, *p* < 0.05); **, very significant difference between the two groups of data (*n* = 3, *p* < 0.01); ***, very significant difference between the two groups (*n* = 3, *p* < 0.001).

**Figure 8 microorganisms-08-02023-f008:**
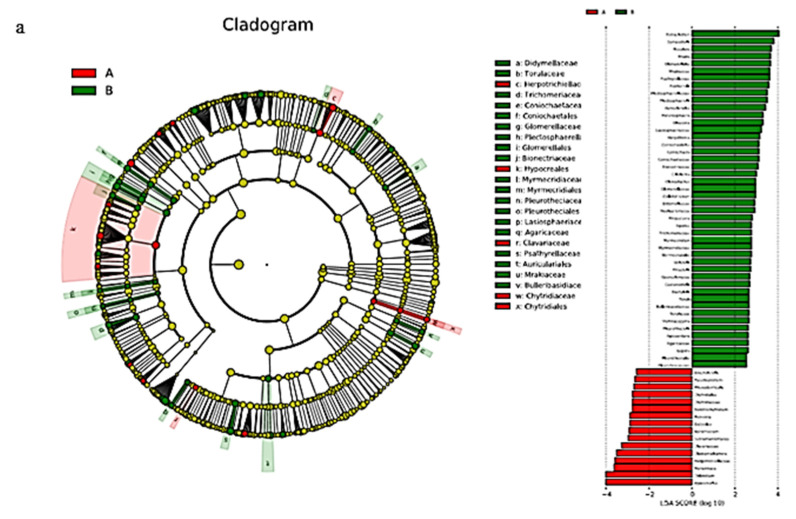
LEfSe analysis of litter-layer fungi. Cluster tree of litter-fungus LEfSe analysis in (**a**) Korean pine plantation forest and original Korean pine forest, (**b**) different litter layers of Korean pine plantation forest, and (**c**) different litter layers of original Korean pine forest.

**Figure 9 microorganisms-08-02023-f009:**
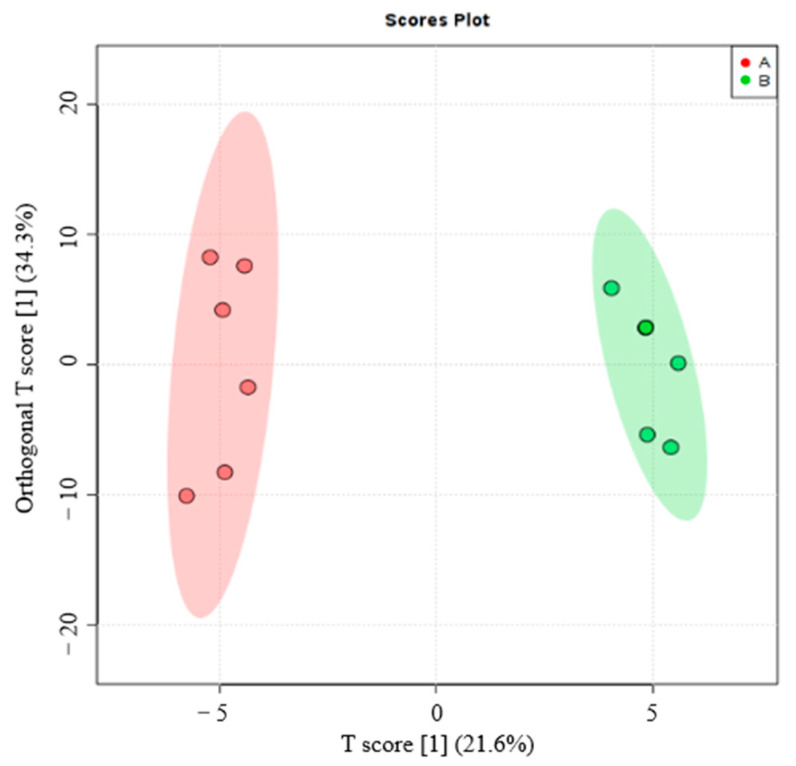
Orthogonal partial least squares discriminant analysis (OPLS–DA) score diagram shows the screening of different litter metabolites of plantation (**A**) and original forest (**B**).

**Figure 10 microorganisms-08-02023-f010:**
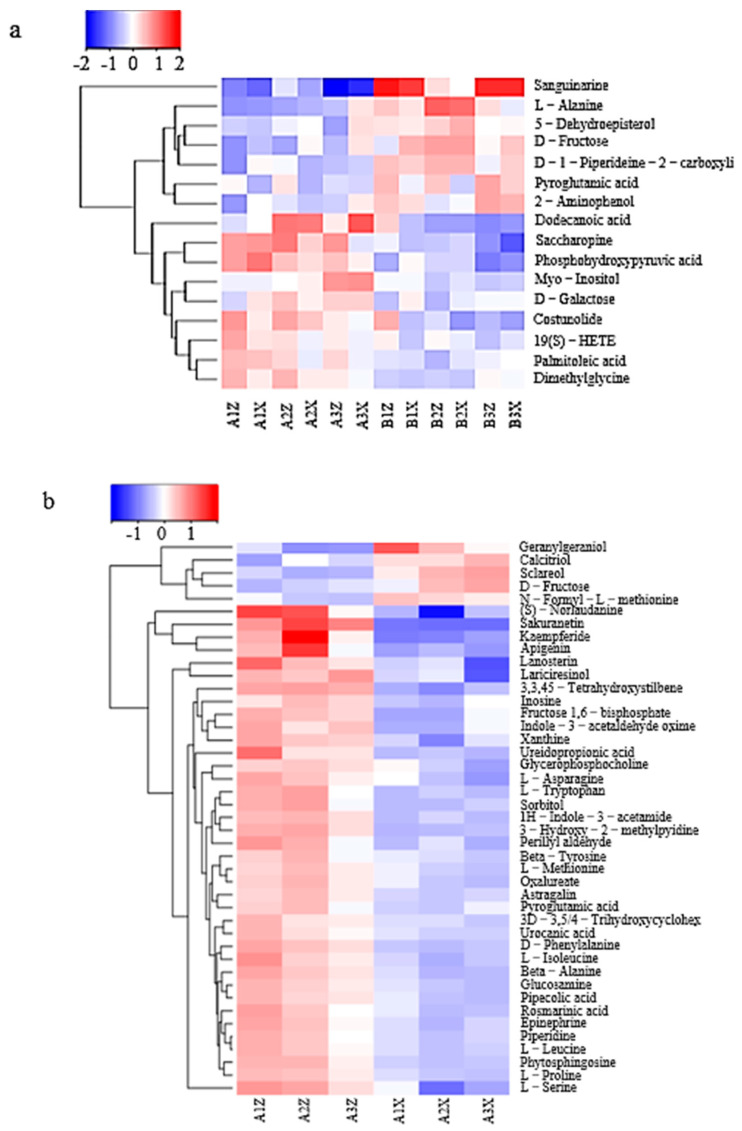
Heatmap analysis of different metabolites in four groups of litter. Clustering heat map of differential metabolite analysis between litter groups of (**a**) original and plantation forest, (**b**) middle and lower layers of plantation forest, and (**c**) middle and lower layers of original forest.

**Figure 11 microorganisms-08-02023-f011:**
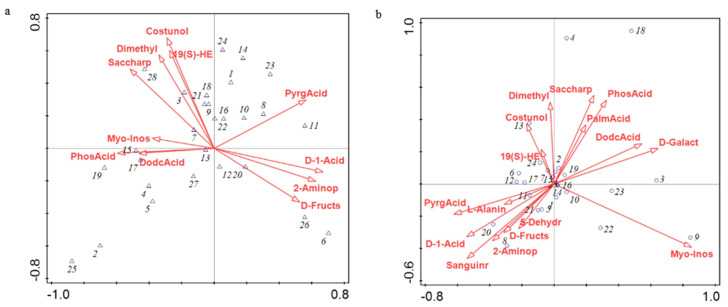
Canonical correspondence analysis of microbial relative abundance and differential metabolites. (**a**) Canonical correspondence analysis of relative abundance and differential metabolites of bacteria. Different numbers represent different bacterium genera: 1, *Acidibacter*; 2, *Acidobacteriales_norank*; 3, *Bradyrhizobium*; 4, *Bryobacter*; 5, *CandidatusSolibacter*; 6, *CandidatusUdaeobacter*; 7, *Chitinophagaceae_uncultured*; 8, *Chthoniobacter*; 9, *Dongia*; 10, *Ferruginibacter*; 11, *Flavobacterium*; 12, *Gemmatimonadaceae_uncultured*; 13, *Haliangium*; 14, *Hirschia*; 15, *Micropepsaceae_uncultured*; 16, *Microscillaceae_uncultured*; 17, *Pedosphaera*; 18, *Phenylobacterium*; 19, *Rhodanobacter*; 20, *SC-I-84_norank*; 21, *SWB02*; 22, *Saccharimonadales_norank*; 23, *Sandaracinaceae_uncultured*; 24, *Sphingomonas*; 25, Subgroup 2_norank; 26, Subgroup 6_norank; 27, *Xanthobacteraceae_uncultured*; 28, *Mucilaginibacter*. (**b**) Canonical correspondence analysis of relative abundance and differential metabolites of fungi. Different numbers represent different genera: 1, *Cercospora*; 2, *Chalara*; 3, *Chloridium*; 4, *Clitocybe*; 5, *Dactylaria*; 6, *Golovinomyces*; 7, *Graphilbum*; 8, *Hymenoscyphus*; 9, *Lactarius*; 10, *Mortierella*; 11, *Mrakia*; 12, *Mycoarthris*; 13, *Mytilinidion*; 14, *Phialea*; 15, *Phialocephala*; 16, *Polyscytalum*; 17, *Pseudoanungitea*; 18, *Russula*; 19, *Sporidesmium*; 20, *Sympodiella*; 21, *Tetracladium*; 22, *Tomentella*; 23, *Trichoderma*; 24, *Xenochalara*; 25, *Xenopolyscytalum*.

**Table 1 microorganisms-08-02023-t001:** Introduction of terraced fields.

Forest	Component Characteristics	Age (Year)	Litter Thickness (cm)
Korean pine plantation	Artificial mixed forest of *Pinus koraiensisSieb. et Zucc., Piceaasperata Mast., Larix gmelinii (Rupr.) Kuzen., Pinus sylvestris var. mongolicaLitv.**,* *Fraxinus mandshuricaRupr.*, and other tree species.	65 (afforestation, 1954)	2–13
Original Korean pine forest	*Pinus koraiensisSieb. et Zucc* is the main species, accompanied by a variety of warm hardwood species (*Tilia tuan Szyszyl., Betula., Quercus mongolica Fisch. ex Ledeb., Ulmus laciniata (Trautv.) Mayr., Acer mono Maxim., Populus ussuriensisKom.*), and by some cold and temperate zones tree species in Eurasian coniferous forests, such as *PiceakoraiensisNakai., Piceajezoensis var. microsperma., Abies nephrolepis (Trautv.) Maxim*, coniferous, broad-leaved mixed forest.	>300	2–15

**Table 2 microorganisms-08-02023-t002:** Bacterial microbial-diversity index of plantation forest litter and original forest litter.

Sample Number	Chao1	Shannon	Simpson
AZ	3733 ± 226 ^c^	6.77 ± 0.22 ^b^	0.0049 ± 0.0032 ^a^
AX	4030 ± 534 ^b^	7.06 ± 0.20 ^a,b^	0.0025 ± 0.0011 ^a,b^
BZ	4853 ± 292 ^a,b^	7.36 ± 0.11 ^a^	0.0015 ± 0.0003 ^b^
BX	5937 ± 795 ^a^	7.53 ± 0.11 ^a^	0.0012 ± 0.0003 ^b^

Note: Data in the table represent three repeated means ± standard deviations. There were significant differences between different letters in different treatments (*p* < 0.05).

**Table 3 microorganisms-08-02023-t003:** Fungal microbial-diversity index of plantation forest and original forest.

Sample Number	Chao1	Shannon	Simpson
AZ	634 ± 125 ^a^	4.09 ± 0.42 ^b^	0.0436 ± 0.019 ^b^
AX	496 ± 96 ^b^	4.01 ± 0.67 ^b^	0.0538 ± 0.027 ^a^
BZ	589 ± 103 ^a,b^	4.34 ± 0.46 ^a^	0.0377 ± 0.024 ^b^
BX	582 ± 152 ^a,b^	4.08 ± 0.43 ^b^	0.0543 ± 0.015 ^a^

Note: Data represent three repeated means ± standard deviations. There were significant differences between different letters in different treatments (*p* < 0.05).

**Table 4 microorganisms-08-02023-t004:** Metabolite statistics of litter difference (variable importance in projection (VIP) ≥ 1 and *p* ≤ 0.05).

Negative	Differential Metabolites	Positive	Differential Metabolites
A and B	7	A and B	16
AZ and AX	22	AZ and AX	43
BZ and BX	9	BZ and BX	21

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
