# Peer review of "Differences in Microbial Community and Metabolites in Litter Layer of Plantation and Original Korean Pine Forests in North Temperate Zone"

_microorganisms, 2020, doi:10.3390/microorganisms8122023_

Round 1
Reviewer 1 Report
Overview
This manuscript describes a study evaluating microbial diversity and related metabolites in the litter of Korean pine forests. This topic is very important and interesting to understanding mechanisms of plant residues decomposition. Unfortunately, it seems to me that this manuscript has flaws that severely limit its value.
First, significant English correction is required. In some sentences there is not possible to understand what authors want to say.
Second, authors hypothesize and made a conclusion that decomposition rate of forest litters is closely related to microbial community composition. However, as we can see from the Methods section and from the Results authors did not measure decomposition rates of forest litter neither in natural Korean pine forests nor in the Korean pine plantations. So this hypothesis cannot be supported by the results of the study.
Line-by-line comments:
Abstract:
Line 16: Which punctuation mark must be in this sentence: full stop or comma?
Line 19: Do you mean LEfSe analysis? Please use correct abbreviation of the method here and through the text of the manuscript.
Line 25: Unfortunately, it is not clear from the Abstract what is Acidobacteria subgroup 6. What is its difference from other subgroups of Acidobacteria. In addition from the sentence on the Lines 245-246 it looks like that Acidobacteria and Subgroup_6 are two different groups of microorganisms.
The following sentence (Lines 25-26) “It can grow with carbon compounds as the substrate and participate in the metabolism of carbon compounds.” is incomprehensible.
Line 33: Used term “material cycle” is common in economic science but not in the natural sciences.
Introduction:
Lines 56-60: What do you mean as a kind: is this a species or a group of species? Please be more specific. "Kind" is a loose colloquial word with no scientific meaning. And what is the “forest-type door”? Please explain this. In total this sentence is incomprehensible.
Materials and Methods:
Line 104-105: Please use bark instead skins. What is the difference between herbs and plants? Residues is the more common word in studies of soil organic matter.
Sentence in the Lines 110-114 is completely not understandable.
Table 1: Latin names of tree species are required. What means warm and cold tree species? Please explain this.
Line 122: Litter is not an organism; it has not its own DNA. You have extracted DNA of microorganisms and fungi living in the litter. Please be more specific.
Line 140-141: Did you really add 100L of water to extract metabolites from 100g of litter and then used 5L of supernatant for chromatography? Is it correct? Maybe it should be 100 and 5 ml?
Results:
Fig.2: What is A and B panels on this figure?
Line 211: Authors write: “Figure 3a shows that there were 41 significant differences between the Korean pine plantation and original Korean pine forest …”. Which differences? Please be more specific and give the detailed description which differences this figure shows.
Line 229: LEfSe instead lefse.
Fig. 6: Please indicate what is A and B panels on this figure.
Line 363: What means lower litter?
Line 384: Which genus correlated influencing factors?
Discussion:
Lines 403-407: This paragraph should be rewritten.
Line 446-447: Cellulose is also organic matter.
Conclusion:
Authors did not study rate of the litter decomposition so it is not correct to conclude that decomposition rate of forest litter was closely related to the composition of microbial communities and related metabolites. Results of the study does not show that metabolites provide acceleration of degradation by microorganisms. It is impossible to compare decomposition rate of litter on Korean pine plantation and in the natural forest because authors did not study decomposition rate of these litters. So the most part of Conclusion is not related to the results obtained.
Author Response
Thank you for your valuable comments. I've uploaded point-by-point replies in word.

Reviewer 2 Report
The authors compared the microbial diversity for both bacteria and fungi between litter layers of original and planted pine forest in Korea. Moreover they tried tom find relationships between decomposition rate in litters and its metabolite composition. It is new to investigate together the microbial diversity and metabolite composition in a pine forest litter layer.
The manuscript contain many new information and approaches, therefore I suggest to accept it with only a minor revisions.
Remarks:
L115: Please spaecify more clearly, that there were two forest types investigated: plantation (A) and original forest(B), within two litter layers middle(X) and lower layer(Z).
L124: …V3-V4 region of the 16S rRNA gene.
L140-141: 100 and 5 L were taken, not mL?
In Figure 1.Title: please specify the X and Z, (AX, AZ; BX, BZ)and also that it shows the OTU numbers of bacteria.
Similarly, in Figure 5 Title please specify that this relate to the OTU numbar of fungi, and the meaning of X and Z.
Figure 9.Title: please specify in more details, what can be see and relates in the graph.(screening of different litter metabolites).
In Appendix A, Figure 1&2: What are the different treatments?
Author Response

(The authors gave the same response as above.)

Round 2
Reviewer 1 Report
Authors improved content of the text and change conclusions so that now it is supported by the obtained results.
However authors did not improve English language of the manuscript so that many questions still arise to the text.
For example, only in the Abstract:
Lines 17 and 19: It is not good to start a sentence from a coordinating conjunction "And".
Line 25: "The two forest types contained 285 kinds of organic compounds ...". I think that authors want to say that forest litter contained ..., but not forest types. If we take all components of these forest types then the number of organic compounds will be much much more.
Line 32: Authors should avoid too long "noun chain".
Line 33-34:" ... than those in the middle and northern temperate forest litter layer ...". It is not clear what authors compared: different litter layers or different temperate forests (middle and northern)?
Line 35-36: "... accelerate the degradation of ... microorganisms."
Is this that authiors realy wanted to say or it is a problem of their English language?
In total the sentense on lines 32-36 is too complicate to understand.
Lines 36-38: " ...participation in the elements of ecosystem."
Maybe you meaned "participation in the element cycles in forest ecosystems"?
The similar questions arise to the main text.
I think that text of the manuscript needs significant English edditing.
Author Response

(The authors gave the same response as above.)
